# Urinary Oxidative Stress Biomarker Levels Might Be Useful in Identifying Functional Bladder Disorders in Women with Frequency and Urgency Syndrome

**DOI:** 10.3390/jcm12062336

**Published:** 2023-03-17

**Authors:** Yuan-Hong Jiang, Jia-Fong Jhang, Hann-Chorng Kuo

**Affiliations:** Department of Urology, Hualien Tzu Chi Hospital, Buddhist Tzu Chi Medical Foundation, Tzu Chi University, Hualien 970, Taiwan

**Keywords:** urine, biomarkers, lower urinary tract dysfunction, bladder

## Abstract

Purpose: lower urinary tract dysfunctions (LUTDs) are difficult to diagnose based on symptoms. This study used a cluster of urinary biomarkers, including inflammatory cytokines, neurogenic proteins, and oxidative stress biomarkers, to identify LUTDs in women with frequency and urgency symptoms. Methods: in total, 253 women with video urodynamics (VUDS)- and cystoscopy-confirmed detrusor overactivity (DO), interstitial cystitis/bladder pain syndrome (IC/BPS), dysfunctional voiding (DV), and hypersensitive bladder (HSB), and normal controls were included. Before diagnosis and treatment, urine samples were collected for analysis of biomarkers. The urine levels of biomarkers were compared between groups with bladder dysfunctions and controls and were combined to test the sensitivity in identifying total pathological bladder diseases and specific bladder diseases. Results: After video urodynamic study, VUDS, and urological examinations, bladder dysfunctions were classified into DO (*n* = 31), IC/BPS (*n* = 114), DV (*n* = 45), HSB (*n* = 29), and control (*n* = 34) groups. By using a cystomeric bladder capacity of ≤350 mL, 186/219 (84.9%) of the patients with DO, IC/BPS, DV, and HSB can be discriminated from the controls. Among these urine biomarkers, oxidative stress biomarkers 8-isoprostane, 8-hydroxydeoxyguanosine (8-OHdG), or total antioxidant capacity (TAC) are useful for identifying pathological bladder dysfunction (DO, IC/BPS, and DV) and HSB. With elevated IL-1β and lower IL-2, and elevated TNF-α levels, most patients with DV can be identified. Between DO and IC/BPS, a higher NGF level can identify 58.3% of IC/BPS cases, whereas a lower NGF level can identify 75.0% of DO cases. Conclusion: by using a cluster of urine biomarkers, DO, IC/BPS, and DV cases can be identified based on elevated levels of urine oxidative stress biomarkers 8-isoprostane, TAC, or 8-OHdG, and HSB cases with a low TAC. These urine biomarkers are useful for identifying specific LUTDs in women with frequency and urgency symptoms.

## 1. Introduction

Lower urinary tract symptoms (LUTS) are common complaints in women. Patients may have storage or voiding symptoms or both. Detrusor overactivity (DO), interstitial cystitis/bladder pain syndrome (IC/BPS), and dysfunctional voiding (DV) are commonly encountered in women with LUTS. Although urgency is the cardinal symptom for DO, patients with IC/BPS and DV-related bladder outlet obstruction (BOO) might also have these complaints [1,2]. In addition, some women may only have a hypersensitive bladder or completely normal lower urinary tract function; however, LUTS might mimic those in functional bladder disorders. With urodynamic study or video urodynamic study and cystoscopic hydrodistention, lower urinary tract dysfunctions (LUTDs) can be identified, and women who can receive different medications [3]. However, most women do not accept these invasive examinations, even for making an accurate diagnosis before the intake of medications. Therefore, a noninvasive, easily performed, and inexpensive diagnostic method must be developed to identify LUTDs or specific bladder disorders in women with LUTS.

Urine contains proteins and is excreted from the kidney or bladder wall. Previous studies have found that some urinary cytokines, chemokines, and oxidative stress biomarkers are elevated in patients with overactive bladder (OAB), DO, IC/BPS, and DV [4,5,6,7,8]. Although urine-based biomarkers may provide valuable diagnostic and prognostic tools for the clinical assessment of LUTDs and treatment guidelines, the urine proteins currently found in women with different bladder disorders varied widely and are poorly reproducible [9]. Variations in urinary biomarker levels might result from the differing severities of the same bladder disorder, and the varying increases in urine biomarker levels might be caused by the different pathophysiology of a similar symptom syndrome. Thus, we hope to find specific urine proteins that can help identify functional bladder disorders or specific LUTDs from nonbladder disorders in women who have similar frequency and urgency symptoms.

In this study, we aimed to measure urinary biomarkers including inflammatory cytokines, neurogenic proteins, and oxidative stress biomarkers, compare the urine levels of a cluster of these biomarkers between LUTDs caused by a bladder pathology and nonbladder pathology, and identify specific bladder diseases based on the elevated levels of urine biomarkers.

## 2. Materials and Methods

In total, 253 women who had undergone video urodynamic study (VUDS) and further urological examinations were included in this study. After thorough VUDS and urological examinations, a final diagnosis, including DO, IC/BPS, DV, hypersensitive bladder (HSB), and normal VUDS tracing (controls), was made. DO, DV, and HSB were diagnosed according to the recommendations of the International Continence Society [10]. The diagnostic criteria for IC/BPS were based on the ESSIC guidelines, with the exclusion of similar diseases [11]. The detailed inclusion and exclusion criteria were similar to those in our previous study [12]. A pelvic examination was performed to exclude the presence of cystocele or uterine prolapse that might lead to symptoms similar to IC/BPS or DV. Women classified as controls were those with stress urinary incontinence, though without other storage or voiding dysfunctions. For diagnosis, urine samples were collected at full bladder in patients who were medication naïve.

VUDS was performed in all patients and controls. VUDS was performed in accordance with the recommendations of the International Continence Society [10]. The VUDS parameters included detrusor pressure (Pdet), maximum flow rate (Qmax), voided volume (VoL), postvoid residual urine volume, first sensation of bladder filling (FSF), full bladder sensation (FS), cystomeric bladder capacity (CBC), bladder compliance, and bladder contractility index.

This study was approved by the Institutional Review Board and Ethics Committee of the Hospital (IRB: 105-25-B, 105-31-A, 107-175-A). All participants had been involved in different clinical trials for the treatment of LUTDs and IC/BPS. Patients had been informed about the rationale of the study, and informed consent was obtained when urine samples were collected in previous clinical trials.

### 2.1. Urinary Biomarker Investigations

Urine cytokine and chemokine investigations were conducted in accordance with our previous study [13]. Briefly, a total of 50 mL of urine samples were collected at an FS before any surgical procedure or medical treatment. Urine samples were obtained by self-voiding, and urine samples with confirmed urinary tract infections were excluded. The urine samples were placed immediately on ice before transferring them to the laboratory. These samples were then centrifuged at 1800 rpm for 10 min at 4 °C. The supernatant was preserved in a freezer at −80 °C. The frozen samples were centrifuged at 12,000 rpm for 15 min at 4 °C before further analyses were performed, and the supernatants were used for subsequent measurements. Urinary proteins were quantified using enzyme-linked immunosorbent assay (ELISA) kits according to the manufacturer’s instructions [12,13]. The proteins and kits included oxidative stress biomarkers such as total antioxidant capacity (TAC) (Abcam, Cambridge, UK), 8-hydroxy-2′-deoxyguanosine (8-OHdG) (BioVision, Waltham, MA, USA), and 8-isoprostane (Enzo Life Science, Farmingdale, NY, USA); inflammatory cytokines, including interleukin (IL)-1β, IL-2, IL-6, IL-8, tumor necrosis factor (TNF)-α, and vascular endothelial growth factor (VEGF) (Millipore, Darmstadt, Germany); and neurogenic proteins including nerve growth factor (NGF), brain-derived neurotrophic factor (BDNF), and prostaglandin E2 (PGE_2_).

Commercial microspheres with the Milliplex^®^ Human Cytokine/Chemokine Magnetic Bead-based Panel Kit (Millipore, Darmstadt, Germany) were used to assay inflammation- and neurogenic-related urinary cytokines and chemokines. According to urinary cytokines and chemokines, which were previously reported to be significant in the diagnoses of DO, IC/BPS, and BOO, nine targeted analytes were selected, including IL-1β, IL-2, IL-6, IL-8, TNF-α, VEGF, NGF, BDNF, and PGE_2_. These analytes were then measured by the multiplex kit (Cat. No. HCYTMAG-60K-PX30). The procedures to measure these urinary cytokines and chemokines were based on the manufacturer’s instructions and as reported previously [12,13]. To calculate the individual cytokine/chemokine levels in the urinary samples, the median fluorescence intensity of each cytokine/chemokine target was recorded and analyzed.

The quantification of 8-OHdG, 8-isoprostane, and TAC in urine samples was performed in accordance with the manufacturer’s instructions (8-OHdG ELISA kit, Biovision; 8-isoprostane ELIZA kit, Enzo; and TAC assay kit, Abcam) as in our previous report [14]. The median fluorescence intensities of the targets were analyzed to calculate the corresponding concentrations in the samples. The measurements of urine 8-isoprostane levels were standardized by urinary creatinine levels.

### 2.2. Statistical Analysis

Continuous variables were expressed as means ± standard deviations and categorical data as numbers and percentages. VUDS parameters were compared among all subgroups, and the levels of urine biomarkers were compared among subgroups. Urine biomarkers were also compared between patients with pathological bladder disorders (DO, IC/BPS, and DV) and those without definite bladder pathologies (HSB and controls). The urine levels of urine biomarkers, which were significantly higher in pathological bladder disorders, were further calculated by the receiver operating characteristics analysis to yield a cutoff value (COV) and area under the curve (AUC). The percentages of patients with urine biomarker levels higher than the COV were calculated in each subgroup. Then, several urine biomarkers that had a sensitivity of >60% were combined to test the sensitivity of these combined biomarkers in identifying total pathological bladder diseases and specific bladder diseases. The IBM SPSS Statistics for Windows version 20.0 (IBM Corp., Armonk, NY, USA) was used for statistical analysis, and *p* values of <0.05 were considered statistically significant.

## 3. Results

After VUDS and urological examinations, the bladder conditions were classified into DO (*n* = 31), IC/BPS (*n* = 114), DV (*n* = 45), HSB (*n* = 29), and control (*n* = 34) subgroups. The VUDS parameters are listed in Table 1. The Pdet was highest in the patients with DV. Patients with IC/BPS, DV, and HSB had a significantly lower Qmax than those with DO and the controls. Bladder FSF and FS were increased in patients with DO, IC/BPS, and DV than in those with HSB and the controls. The CBC was smaller in patients with DO, IC/BPS, DV, and HSB than in controls. Patients with DO, IC/BPS, and DV had lower bladder compliance than controls.

Table 2 shows urine biomarkers among all subgroups and controls. Significantly higher urinary levels of 8-isoprostane and TAC were noted in patients with DO and IC/BPS. The 8-OHdG levels were significantly higher in patients with IC/BPS and DV, though not in those with DO. In this study, urine levels of IL-1β and TNF-α were significantly higher only in patients with DV; however, IL-2 was significantly lower than that in other subgroups. By contrast, the VEGF levels were significantly higher in patients with DO and IC/BPS than in those with DV and HSB. When DO, IC/BPS, and DV were grouped as pathological bladder diseases, 8-isoprostane, TAC, 8-OHdG, VEGF, NGF, and PGE_2_ were significantly higher, though IL-2 was significantly lower than in those with HSB, and the controls (Table 3). By using the AUC and COV of each urine biomarker to discriminate patients with DO, IC/BPS, and DV versus patients with HSB and controls, only TAC (≥844.3) and 8-OHdG (≥24.13) had an AUC > 0.70. Other urine biomarkers did not reach a satisfactory AUC using the COV.

When the correlation between urine biomarker levels and VUDS parameters was analyzed, a significant association was found between 8-isoprostane and volume (−0.145, *p* = 0.024) and bladder compliance (−0.130, *p* = 0.043), a significant correlation between TAC and bladder compliance (−0.162, *p* = 0.011), and a significant correlation between 8-OHDG and Qmax (−0.143, *p* = 0.025), Vol (−0.238, *p* = 0.000), FS (−0.194, *p* = 0.002), and CBC (−0.228, *p* = 0.000). The COVs of these significant urine biomarkers and the sensitivity of each urine biomarker and combined urine biomarkers for each bladder disease are shown in Table 4. From the percentage of patients with COV values higher in these urine biomarkers, patients with DO, IC/BPS, and DV had a higher 8-isoprostane or TAC (DO and IC/BPS) or a higher 8-OHdG level (IC/BPS and DV) than patients with HSB. Patients with DV had higher IL-1β and TNF-α, and lower IL-2 levels. In addition, patients with DO and IC/BPS had a higher rate, whereas patients with DV had a lower rate of having higher VEGF levels. Among all subgroups, patients with IC/BPS had higher NGF levels. Based on the above differences in urine biomarker levels, we tried to test the AUC, sensitivity, and specificity by a urine biomarker level in combination with two or three bladder dysfunction subgroups. As shown in Table 5, 8-isoprostane, TAC, 8-OHdG, IL-1β, IL-2, TNF-α, and VEGF had an AUC of >0.700 and good sensitivity and specificity for the given single or combined bladder dysfunction subgroups.

Figure 1 shows the diagnostic algorithm using urinary biomarkers to identify LUTDs in women with LUTS. By using a CBC of ≤350 mL, 186/219 (84.9%) of the patients with DO (25/31, 80.6%), IC/BPS (90/114, 78.9%), DV (82.2%), and HSB (25/29, 86.2%) can be discriminated from the controls. Among these patients, a lower TAC level can be used to identify HSB (21/25, 84.0%), leaving patients with DO, IC/BPS, and DV. If the IL-1β levels were elevated, IL-2 (21/27, 77.8%) lower, and TNF-α (21/21, 100%) elevated, most of the patients with DV can be identified. If the 8-isoprostane or 8-OHdG or TAC values were higher, we can further identify 87.6% of patients with DO (20/25, 80%), IC/BPS (94/99, 94.9%), and DV (27/37, 73.0%). Then, based on an elevated VEGF, 80% of patients with DO (20/20, 100%) and IC/BPS (72/94, 76.6%) can be separated from those with DV (23/27, 85.2%). Finally, between DO and IC/BPS, a higher NGF level can be identified in 58.3% of patients with IC/BPS (42/72), whereas a lower NGF level can be identified in 75.0% of DO cases (15/20).

## 4. Discussion

The results of this study revealed that bladder dysfunction in women with LUTS can be discriminated against using clusters of urinary levels of inflammatory proteins, neurogenic proteins, and oxidative stress biomarkers. Among these urine biomarkers, oxidative stress biomarkers 8-isoprostane, 8-OhdG, and TAC are useful for identifying pathological bladder dysfunctions (DO, IC/BPS, and DV) and HSB. Increased levels of IL-1β and TNF-α can be used to identify DV, whereas an increased NGF level can be used to distinguish IC/BPS and DO.

Urinary proteins had long been considered to have a diagnostic role in bladder dysfunction or bladder outlet dysfunctions [4,5,7,8]. However, owing to variations in urine collection, different renal conditions, and medical comorbidities, urine levels of biomarkers varied widely in the studies of OAB, IC/BPS, and DV-related BOO [9]. A single urinary protein level might not have good sensitivity and specificity for a certain LUTD; however, if serial urinary proteins and biomarkers can be used, a certain bladder dysfunction might be identified with satisfactory sensitivity and specificity in patients with specific clinical characteristics or urinary biomarker levels.

A high degree of overlapping was found between symptoms of LUTDs in women with frequency urgency and dysuria [1,2]. Accurate identification of specific LUTD is not easy based on the clinical symptoms, voiding diary, and uroflowmetry results. Therefore, an effective treatment might not be achieved without a VUDS examination. With the aid of urinary biomarker levels, we might treat patients under a more accurate diagnosis with a more rational pathophysiology basis.

Any LUTD has specific pathophysiology and clinical symptoms. Patients with DO usually have increased bladder sensation, small CBC, involuntary detrusor contractions during the storage phase, and involuntary detrusor contraction at bladder capacity. These bladder dysfunctions result in urgency, frequency, and urge urinary incontinence (UUI) [15]. Patients with IC/BPS also have increased bladder sensation; however, involuntary detrusor contractions are absent. Therefore, patients usually do not have urgency or UUI, though bladder pain might occur with a full bladder [16]. A previous investigation revealed an important correlation between IC/BPS and rheumatic, autoimmune, and chronic inflammatory diseases. These disorders may share some pathophysiologic mechanisms [17]. Patients with DV also have frequency and urgency symptoms due to BOO. Therefore, difficult urination and incomplete bladder emptying are common. Some patients with DV also have DO during storage or at bladder capacity because of increased oxidative stress [18]. Patients with HSB might have increased bladder sensation caused by bladder inflammation or psychosomatic dysfunction, causing frequency and nocturia, but not urgency, UUI, or bladder pain. Chronic stress could lead to chronic systemic inflammation and result in LUTS [19]. These patients could be carefully identified or discriminated through their symptoms and a voiding diary. However, in real-life practice, patients usually cannot clearly report their symptoms and might be confused with the definition of LUTS [2]. A cluster of urinary biomarkers might provide objective evidence for the underlying pathophysiology and lead physicians to give initial treatment based on the urine levels of these urine biomarkers.

In recent decades, despite enthusiasm for the search for significant urine biomarkers to make a diagnosis, differentiate among LUTDs, and predict treatment outcomes, we still cannot have a conclusive result to obtain informative and reproducible urinary biomarkers for discriminating OAB, IC/BPS, and BOO [20,21,22]. The major deficiency of poor reproducibility might be the unclear classification of LUTDs and the lack of clinical covariate adjustment [9]. This study used VUDS and cystoscopic hydrodistention to diagnose HSB, DO, IC/BPS, and DV in these women with LUTS, and the results should be more accurate in the diagnosis than previously reported data.

According to these data, since most women with bladder dysfunctions had a smaller CBC than the controls, we could use a small CBC (<350 mL) as the first biomarker to exclude most of the women with normal bladders, leaving patients with DO, IC/BPS, DV, and HSB. In addition, since oxidative stress biomarkers 8-isoprostane and TAC were significantly increased in women with DO and IC/BPS and the 8-OHdG level was significantly elevated in women with IC/BPS and DV, the elevation of either one of these three oxidative stress biomarkers could identify 87.6% of women with DV, whereas the use of a lower urine level of TAC could identify 84.0% of women with HSB. The remaining 141 women with DO, IC/BPS, and DV could be further discriminated by an elevated urine VEGF level for 80.7% of women with DO and IC/BPS or an elevated IL-1β level or TNF-α for 77.8% of women with DV. Finally, a lower urine NGF level could be used to separate DO and a higher NGF level for IC/BPS. Although not all women with bladder dysfunction could be identified, by using this diagnostic algorithm by CBC and different urine biomarker levels, we still could discriminate different bladder dysfunctions in women with frequency and urgency syndrome.

Among the urine levels of inflammatory proteins, neurogenic proteins, and oxidative stress biomarkers, oxidative stress biomarkers are the most valuable to discriminate DO, IC/BPS, and DV from HSB and normal controls. Pelvic ischemia and oxidative stress have been considered to play important roles in LUTDs such as DO, OAB, and IC/BPS [23]. In atherosclerosis-induced bladder ischemia, oxidative stress and inflammation are found to result in DO in animal models and humans [24,25]. Increased urine levels of 8-OHdG and 8-isoprostane have been noted in patients with IC/BPS, OAB, and DV [14,21,26]. Significantly decreased bladder perfusion during filling and at capacity was confirmed in patients with IC/BPS [27,28]. Increased expression levels of hypoxia-inducible factor-1α, VEGF, and immature vascularization in the bladder tissue of patients with IC/BPS were further identified [29,30]. Based on this evidence, we might identify IC/BPS by the elevated oxidative stress biomarkers and elevated VEGF levels. In this study, patients with DO, IC/BPS, and DV, though not HSB and normal controls, had higher urine levels of 8-isoprostane, TAC, or 8-OHdG. Therefore, it is feasible to identify these bladder dysfunctions based on oxidative stress biomarkers.

Previous studies have shown that cytokines and chemokines play important roles in the pathogenesis of several chronic inflammatory diseases [12,13,20,21,22]. These cytokines may have crosstalk with the nervous system, resulting in the hypersensitization of pain receptors and pain via neurogenic inflammation [31,32,33]. Hence, urine biomarkers could represent the pathophysiological mechanism of bladder pathologies or bladder outlet dysfunctions, and these LUTDs could be identified by urine levels of different biomarkers [13,34]. Changes in urine biomarker levels can be also used to assess therapeutic outcomes after specific treatments [35]. However, different LUTDs might have the same elevated urine levels of a single urine biomarker, whereas an LUTD could have different urine biomarker levels due to varying disease severity. Therefore, the use of a cluster of urinary biomarker levels might allow us to identify patients with different LUTDs and those without bladder-centered pathophysiology.

In this study, we found that oxidative stress biomarkers can be the most useful tools to identify patients with bladder pathologies such as DO, IC/BPS, and DV. We can also identify patients with HSB based on a lower urine TAC level, suggesting that patients with HSB may have low oxidative stress and a lower antioxidative capacity. In these patients with LUTD caused by bladder pathologies, increases in bladder inflammation and intravesical pressure or a decrease in blood supply might contribute to the increased oxidative stress [36,37,38]. Since high urine levels of IL-1β and TNF-α were noted in patients with DV, patients with DV can be further differentiated from those with DO and IC.

The results of this study revealed that oxidative stress and inflammation are important factors contributing to LUTDs due to different bladder pathologies. Tissue hypoxia and inflammation could generate reactive oxygen species (ROS), which serve as mediators in both physiological and pathological signaling transductions [39]. Excessive ROS production can result in oxidative stress, which leads to damage in cellular DNA, lipids, and proteins, resulting in changes in the structure and function of target tissues with a pathological link in many diseases including LUTDs [40,41]. By using clusters of urine oxidative stress biomarkers and inflammatory and neurogenic cytokines, different LUTDs could be identified in women with similar clinical symptoms of frequency and urgency.

Although this study did not provide a single urine biomarker to identify one lower urinary tract dysfunction, we still can make a differential diagnosis by the cluster of biomarkers, step by step, to exclude normal female patients (by a voided volume >350 mL) and identify patients with hypersensitive bladder (by a lower TAC level). For the patients with elevated 8-isoprostane, 8-OHdG, and TAC, DO, IC/BPS, and DV should be considered. Based on a lower VEGF and elevated IL-1β, DV can be identified, leaving patients with DO and IC/BPS. Between them, a lower NGF level favors the presence of DO, whereas a higher NGF level predicts IC/BPS. In real-life practice, we can add the clinical symptoms to improve the predictive rate, such as urgency and urgent urinary incontinence for DO, bladder pain for IC/BPS, dysuria, and a low maximum flow rate for DV.

This study has some limitations. First, no comorbidity control was made in all LUTD subgroups since systemic diseases might have chronic inflammation that also increases levels of urine biomarkers. Second, we collected urine samples at a full bladder, however, the bladder volume was not well controlled, which might result in bias in the urine biomarker density. Nevertheless, the results of this study provide evidence that the urine levels of a cluster of urine biomarkers could identify a satisfactory proportion of patients with specific LUTDs such as DO, IC/BPS, DV, and HSB from women who have frequency and urgency symptoms. Further studies with a well-controlled patient cohort are necessary.

## 5. Conclusions

The use of a cluster of urine biomarkers allows us to identify patients with DO, IC/BPS, and DV based on elevated urine levels of oxidative stress biomarkers 8-isoprostane, TAC, or 8-OHdG and identify HSB with a low TAC. By an elevated level of IL-1β or TNF-α, DV can be separated from these bladder disorders. An increased VEGF level is associated with DO and IC/BPS, whereas an increased NGF level might further identify IC/BPS. These urine biomarkers are useful parameters for urologists to identify LUTDs in women with frequency and urgency symptoms.

## Figures and Tables

**Figure 1 jcm-12-02336-f001:**
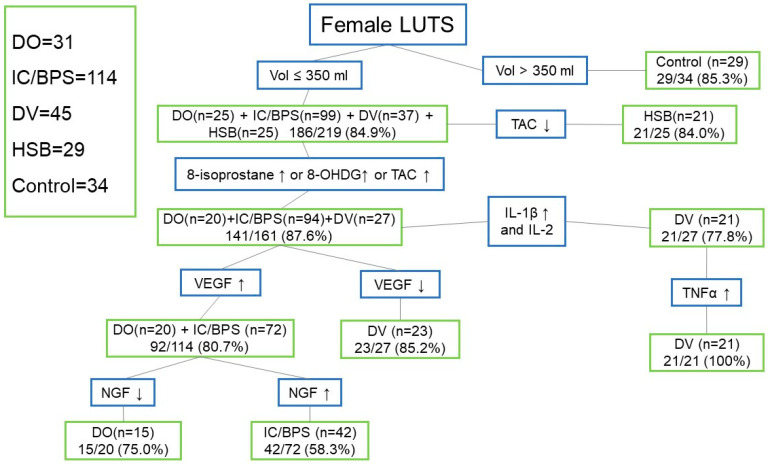
Diagnostic algorithm using urinary biomarkers to identify lower urinary tract dysfunctions in women with lower urinary tract symptoms (Abbreviations: LUTS: lower urinary tract symptoms, Vol: volume, DO: detrusor overactivity, IC/BPS: interstitial cystitis/bladder pain syndrome, DV: dysfunctional voiding, HSB: hypersensitive bladder, TAC: total antioxidant capacity, 8-OHdG: 8-hydroxydeoxyguanosine, IL: interleukin, TNF-α: tumor necrosis factor-α, VEGF: vascular endothelial growth factor, NGF: nerve growth factor, BDNF: brain-derived neurotrophic factor, PGE2: prostaglandin E2).

**Table 1 jcm-12-02336-t001:** Urodynamic parameters of women with bladder dysfunctions and normal tracing.

Urine Cytokines	(A) DO N = 31	(B) IC/BPS N = 114	(C) DV N = 45	(D) HSB N = 29	(E) Normal N = 34	*p* Value ^#^	Post Hoc ^$^
Pdet (cmH_2_O)	18.0 ± 11.0	22.3 ± 17.6	47.8 ± 42.7	11.5 ± 9.49	15.1 ± 7.12	<0.001	ABDE v C; B v DE
Qmax (mL/s)	16.1 ± 7.35	10.3 ± 6.41	10.6 ± 6.78	11 ± 6.69	18.9 ± 8.44	<0.001	AE v BCD,
Volume (mL)	272 ± 134	212 ± 117	229 ± 116	232 ± 126	416 ± 152	<0.001	ABCD v E; A v B
PVR (mL)	14.7 ± 40.8	50.8 ± 103	56.4 ± 66	75 ± 105	17.1 ± 71.9	0.012	A v BC
FSF (mL)	109 ± 48.6	131 ± 59.9	125 ± 55.2	185 ± 207	170 ± 65.8	0.002	ABC v DE,
FS (mL)	172 ± 74.2	201 ± 86.1	200 ± 82.5	243 ± 58.6	293 ± 95.8	<0.001	ABC v DE, D v E
CBC (mL)	286 ± 134.7	251.7 ± 138	279 ± 134	302 ± 96	407 ± 166	<0.001	ABCD v E
Compliance	61.8 ± 42.9	70.3 ± 52.9	75.9 ± 83.8	98.4 ± 84.3	160 ± 101	<0.001	ABC v E
BCI	98.1 ± 36.4	70.5 ± 36.3	97.6 ± 51.5	65.3 ± 37.2	98.7 ± 50.4	<0.001	BD v ACE

#: statistical analysis among all subgroups, ^$^: significant difference in post hoc analysis among subgroups. Abbreviations: DO: detrusor overactivity, IC/BPS: interstitial cystitis/bladder pain syndrome, DV: dysfunctional voiding, HSB: hypersensitive bladder, Pdet: detrusor pressure, Qmax: maximum flow rate, PVR: post-void residual, FSF: first sensation of filling, FS: full sensation, CBC: cystometric bladder capacity, BCI: bladder contractility index. Correlation between 8-isoprostane and Volume (−0.145, *p* = 0.024), compliance −0.130, *p* = 0.043); Correlation between TAC and compliance (−0.162, *p* = 0.011); Correlation between 8-OHDG and Qmax (−0.143, *p* = 0.025), Volume (−0.238, *p* = 0.000), FS (−0.194, *p* = 0.002), CBC (−0.228, *p* = 0.000).

**Table 2 jcm-12-02336-t002:** Urinary biomarker levels in women with different bladder dysfunctions and normal tracing.

Urine Cytokines	(A) DO (N = 31)	(B) IC/BPS (N = 114)	(C) DV (N = 45)	(D) HSB (N = 29)	(E) Control (N = 34)	*p*-Value	Post Hoc ^$^
Age	63.9 ± 8.96	54.6 ± 12.4	53.2 ± 4.2	63.0 ± 11.2	59.8 ± 11.1	<0.001	AD v BC
8-isoprostane	32.5 ± 29.8	39.1 ± 29.6	12.9 ± 14.7	22.8 ± 17.3	17.5 ± 15.5	<0.001	A v C, B v CDE
TAC	1559 ± 1359	1658 ± 1190	604 ± 420	388 ± 279	1107 ± 1017	<0.001	AB v CD, D v E
8-OHDG	26.0 ± 17.7	33.2 ± 17.9	32.4 ± 19.4	18.4 ± 16.6	17.7 ± 13.6	<0.001	A v B, BC v DE
IL-1β	0.61 ± 0.54	0.64 ± 0.49	1.16 ± 1.4	0.71 ± 0.63	0.56 ± 0.26	0.001	ABDE v C
IL-2	0.74 ± 0.19	0.76 ± 0.18	0.28 ± 0.22	0.64 ± 0.14	0.79 ± 0.19	<0.001	ABDE v C, D v BE
IL-6	2.05 ± 2.62	1.72 ± 1.53	2.14 ± 5.16	1.53 ± 1.71	1.22 ± 1.29	0.582	
IL-8	20.7 ± 34.4	14.2 ± 15.8	31.0 ± 63.9	48.3 ± 97.7	13.6 ± 22.8	0.060	
TNF-α	0.87 ± 0.4	0.78 ± 0.42	1.21 ± 0.33	0.92 ± 0.56	0.79 ± 0.31	<0.001	ABDE v C
VEGF	14.6 ± 5.96	14.4 ± 6.81	5.56 ± 4.91	8.44 ± 7.84	11.2 ± 5.3	<0.001	AB v CD, C v E
NGF	0.26 ± 0.07	0.37 ± 0.17	0.21 ± 0.05	0.22 ± 0.07	0.27 ± 0.07	<0.001	A v BC, B v CDE
BDNF	0.6 ± 0.22	0.5 ± 0.17	0.63 ± 0.15	0.61 ± 0.29	0.57 ± 0.14	0.004	B v C
PGE_2_	262 ± 175	239 ± 168	218 ± 187	283 ± 259	171 ± 107	0.077	

^$^: significant difference in post hoc analysis among subgroups. MILLIPLEX^®^map kit (magnetic bead-based panel kit) (commercially available). Abbreviations: DO: detrusor overactivity, IC/BPS: interstitial cystitis/bladder pain syndrome, DV: dysfunctional voiding, HSB: hypersensitive bladder, TAC: total antioxidant capacity, 8-OHdG: 8-hydroxydeoxyguanosine, IL: interleukin, TNF-α: tumor necrosis factor-α, VEGF: vascular endothelial growth factor, NGF: nerve growth factor, BDNF: brain-derived neurotrophic factor, PGE_2_: prostaglandin E2.

**Table 3 jcm-12-02336-t003:** Urine biomarkers in women with DO, IC/BPS, and DV compared to women with HSB and normal tracing.

Urine Biomarkers	DO + IC + DV (N = 190)	HSB + Normal (N = 63)	*p* Value	Cut-Off Value	AUC
8-isoprostane	31.8 ± 28.8	20.0 ± 16.4	<0.001	≥20.8	0.610
TAC	1396 ± 1175	776 ± 846	<0.001	≥844.3	0.704
8-OHDG	31.8 ± 18.3	18.0 ± 14.9	<0.001	≥24.13	0.719
IL-1β	0.76 ± 0.84	0.62± 0.47	0.245	≥0.645	0.592
IL-2	0.64 ± 0.28	0.72 ± 0.18	0.015	≤0.39	0.549
IL-6	1.87 ± 2.94	1.36 ± 1.49	0.194	≥0.825	0.575
IL-8	19.2 ± 36.5	29.5 ± 69.8	0.272	≥1.87	0.587
TNF-α	0.89 ± 0.43	0.85 ± 0.44	0.520	≥1.045	0.531
VEGF	12.4 ± 7.30	9.96 ± 6.66	0.022	≥11.24	0.599
NGF	0.31 ± 0.15	0.25 ± 0.08	<0.001	≥0.315	0.642
BDNF	0.55 ± 0.18	0.59 ± 0.22	0.210	≤0.315	0.524
PGE_2_	236 ± 182	171 ± 107	0.022	≥173.1	0.573

Abbreviations: DO: detrusor overactivity, IC/BPS: interstitial cystitis/bladder pain syndrome, DV: dysfunctional voiding, HSB: hypersensitive bladder, TAC: total antioxidant capacity, 8-OHdG: 8-hydroxydeoxyguanosine, IL: interleukin, TNF-α: tumor necrosis factor-α, VEGF: vascular endothelial growth factor, NGF: nerve growth factor, BDNF: brain-derived neurotrophic factor, PGE_2_: prostaglandin E2.

**Table 4 jcm-12-02336-t004:** Receiver operating characteristics curves and cutoff values of urine biomarkers in women with different bladder dysfunctions and normal tracing.

Urine Biomarker	Cut-Off Value	DO (N = 31)	IC/BPS (N = 114)	DV (N = 45)	HSB (N = 29)	Normal (N = 34)
8-isoprostane	≥19.08	19 (61.3%)	80 (70.2%)	8 (17.8%)	12 (41.4%)	11 (32.4%)
TAC	≥592.2	23 (74.2%)	96 (84.2%)	16 (35.6%)	6 (20.7%)	23 (67.6%)
8-OHDG	≥24.13	16 (51.6%)	77 (67.5%)	31 (68.9%)	7 (24.1%)	9 (26.5%)
IL-1β	≥0.615	9 (29.0%)	31 (27.2%)	39 (86.7%)	7 (24.1%)	11 (32.4%)
IL-2	≤0.39	0	0	34 (75.6%)	0	0
TNF-α	≥1.115	6 (19.4%)	14 (12.3%)	35 (77.8%)	5 (17.2%)	5 (14.7%)
VEGF	≥9.08	27 (87.1%)	87 (76.3%)	6 (13.3%)	11 (37.9%)	20 (58.8%)
NGF	≥0.315	9 (29.0%)	63 (55.3%)	4 (8.9%)	3 (10.3%)	7 (20.6%)
PGE_2_	≥173.1050	19 (61.3%)	66 (57.9%)	18 (40.0%)	14 (48.3%)	9 (26.5%)

Abbreviations: DO: detrusor overactivity, IC/BPS: interstitial cystitis/bladder pain syndrome, DV: dysfunctional voiding, HSB: hypersensitive bladder, TAC: total antioxidant capacity, 8-OHdG: 8-hydroxydeoxyguanosine, IL: interleukin, TNF-α: tumor necrosis factor-α, VEGF: vascular endothelial growth factor, NGF: nerve growth factor, PGE_2_: prostaglandin E2.

**Table 5 jcm-12-02336-t005:** Cutoff value to discriminate bladder dysfunctions and the sensitivity and specificity to identify bladder dysfunctions.

Urine Biomarker	Bladder Disorders	Cut-Off Value	AUC	Sensitivity	Specificity
8-isoprostane	DO + IC	≥19.08	0.751	67.8%	72.6%
TAC	DV + HSB	≤592.2	0.818	72.2%	79.0%
8-OHDG	DO + IC + DV	≥24.13	0.719	65.3%	74.6%
IL-1β	DV	≥0.615	0.837	86.4%	74.6%
IL-2	DV	≤0.39	0.925	75.6%	100%
TNF-α	DV	≥1.115	0.837	76.7%	88.1%
VEGF	DO + IC	≥9.08	0.768	78.5%	67.0%
NGF	DO + IC + DV	≥0.315	0.642	38.4%	84.1%
PGE_2_	DO + IC + DV	≥173.1	0.573	53.2%	64.5%

Abbreviations: DO: detrusor overactivity, DV: dysfunctional voiding, HSB: hypersensitive bladder, TAC: total antioxidant capacity, 8-OHdG: 8-hydroxydeoxyguanosine, IL: interleukin, TNF-α: tumor necrosis factor-α, VEGF: vascular endothelial growth factor, NGF: nerve growth factor, PGE_2_: prostaglandin E2.

## Data Availability

Data used in this study are available upon request from the corresponding author.

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
