# Peer review of "Urinary Oxidative Stress Biomarker Levels Might Be Useful in Identifying Functional Bladder Disorders in Women with Frequency and Urgency Syndrome"

_jcm, 2023, doi:10.3390/jcm12062336_

Round 1
Reviewer 1 Report
COMMENTS TO JCM-2232574
The article deals with an important problem, such as chronic functional disorders of the bladder in women. The methodology of the article is correct. The following corrections are suggested:
1.-A weakness in the article is that several different diseases are mixed together, and precise concrete conclusions are not made for each of the 4 groups of diseases identified. This aspect should be commented on in the discussion.
2.- In line 265, when referring to painful bladder, this phrase and this reference should be included: "The pathophysiological relationship of painful bladder with rheumatic and chronic inflammatory diseases has been investigated" REFERENCE: Lorenzo Gomez, M. F. and S. Gomez Castro
(2004). "Physiopathologic relationship between interstitial cystitis and rheumatic, autoimmune, and chronic inflammatory diseases." Arch Esp Urol 57(1): 25-34.
Author Response
Reviewer #1:
The article deals with an important problem, such as chronic functional disorders of the bladder in women. The methodology of the article is correct. The following corrections are suggested:
1.-A weakness in the article is that several different diseases are mixed together, and precise concrete conclusions are not made for each of the 4 groups of diseases identified. This aspect should be commented on in the discussion.
Reply: Thank you for the comment. As we have mentioned in the first paragraph of Introduction, the four subgroups (HSB, DO, DV, IC/BPS) in women have similar clinical lower urinary tract symptoms. It is not easy to identify one from the others based on symptoms and simple office-based examinations. (Lines 34-45) The aim of this study is to search for a non-invasive, easily performed, and inexpensive diagnostic method must be developed to identify LUTDs or specific bladder disorders in women with LUTS. (Lines 45-47) Although this study did not provide a single urine biomarker to identify one lower urinary tract dysfunction, we still can make differential diagnosis by the cluster of biomarkers, step by step, to exclude normal female patients (by a voided volume > 350ml) and identify patients with hypersensitive bladder (by a lower TAC level). For the patients with elevated 8-isoprostane, 8-OHdG, and TAC, DO, IC/BPS, and DV should be considered. Based on a lower VEGF and elevated IL-1β, DV can be identified, and leaving patients with DO and IC/BPS. Between them a lower NGF level favors the presence of DO, whereas a higher NGF level predicts IC/BPS. In real life practice, we can add the clinical symptoms to improve the predictive rate, such as urgency and urgency urinary incontinence for DO, bladder pain for IC/BPS, dysuria and a low maximum flow rate for DV. We have added this paragraph to the discussion section. (Lines 343- 352)
2.- In line 265, when referring to painful bladder, this phrase and this reference should be included: "The pathophysiological relationship of painful bladder with rheumatic and chronic inflammatory diseases has been investigated" REFERENCE: Lorenzo Gomez, M. F. and S. Gomez Castro (2004). "Physiopathologic relationship between interstitial cystitis and rheumatic, autoimmune, and chronic inflammatory diseases." Arch Esp Urol 57(1): 25-34.
Reply: Thank you for the comment. We have added the statement and this important reference. (Lines 260-262)

Reviewer 2 Report
The study included 253 women with different bladder dysfunctions and normal controls, and urine samples were collected to measure levels of various biomarkers, including inflammatory cytokines, neurogenic proteins, and oxidative stress biomarkers. The results showed that a combination of these urine biomarkers can be useful for identifying specific LUTDs in women with frequency and urgency symptoms, and can discriminate between pathological bladder dysfunctions and normal controls. The study concluded that the use of a cluster of urine biomarkers may help diagnose LUTDs, which can be challenging based solely on symptoms. The development of simple and useful biomarkers is very important in real clinical practice, and I believe this paper is very valuable in this regard.
1. The abstract is too long. According to the submission rules of this medical journal, isn't the maximum number of words in an abstract about 200? Please check.
2. Materials and methods
Why were only female patients included in the study? Why were male patients not included?
3. Materials and methods
Did you check for the presence or absence of pelvic organ prolapse?
4. With respect to IC/BPS, the presence of Hunner's lesion can significantly change the nature of the urine, what are your thoughts on that?
5. Results. lines 144-145
Were there any patients who had DO and DV combined, i.e., DHIC? And if there were such patients, how were they classified?
Author Response

(The authors gave the same response as above.)

Round 2
Reviewer 2 Report
I think the authors have answered my question precisely.